# Amelioration of Metal-Induced Cellular Stress by α-Lipoic Acid and Dihydrolipoic Acid through Antioxidative Effects in PC12 Cells and Caco-2 Cells

**DOI:** 10.3390/ijerph18042126

**Published:** 2021-02-22

**Authors:** Kaniz Fatima Binte Hossain, Mahmuda Akter, Md. Mostafizur Rahman, Md. Tajuddin Sikder, Md. Shiblur Rahaman, Shojiro Yamasaki, Goh Kimura, Tomomi Tomihara, Masaaki Kurasaki, Takeshi Saito

**Affiliations:** 1Graduate School of Environmental Science, Hokkaido University, Sapporo 060-0810, Japan; rumana@ees.hokudai.ac.jp (K.F.B.H.); aktermahmuda.bd@gmail.com (M.A.); shiblu.ju@gmail.com (M.S.R.); kura@ees.hokudai.ac.jp (M.K.); 2Department of Environmental Sciences, Jahangirnagar University, Savar 1342, Bangladesh; 3Department of Public Health and Informatics, Jahangirnagar University, Savar 1342, Bangladesh; 4Faculty of Health Sciences, Hokkaido University, Sapporo 060-0810, Japan; sy0618@eis.hokudai.ac.jp (S.Y.); rui18happy@gmail.com (G.K.); tomitomo1121@gmail.com (T.T.); taksaito@med.hokudai.ac.jp (T.S.); 5Faculty of Environmental Earth Science, Hokkaido University, Sapporo 060-0810, Japan

**Keywords:** glutathione, oxidative stress, reactive oxygen species (ROS), Nrf2, cleaved PARP-1, Akt

## Abstract

α-Lipoic acid (ALA) and its reduced form dihydrolipoic acid (DHLA) are endogenous dithiol compounds with significant antioxidant properties, both of which have the potential to detoxify cells. In this study, ALA (250 μM) and DHLA (50 μM) were applied to reduce metal (As, Cd, and Pb)-induced toxicity in PC12 and Caco-2 cells as simultaneous exposure. Both significantly decreased Cd (5 μM)-, As (5 μM)-, and Pb (5 μM)-induced cell death. Subsequently, both ALA and DHLA restored cell membrane integrity and intracellular glutathione (GSH) levels, which were affected by metal-induced toxicity. In addition, DHLA protected PC12 cells from metal-induced DNA damage upon co-exposure to metals. Furthermore, ALA and DHLA upregulated the expression of survival-related proteins mTOR (mammalian target of rapamycin), Akt (protein kinase B), and Nrf2 (nuclear factor erythroid 2-related factor 2) in PC12 cells, which were previously downregulated by metal exposure. In contrast, in Caco-2 cells, upon co-exposure to metals and ALA, Nrf2 was upregulated and cleaved PARP-1 (poly (ADP-ribose) polymerase-1) was downregulated. These findings suggest that ALA and DHLA can counterbalance the toxic effects of metals. The protection of ALA or DHLA against metal toxicity may be largely due to an enhancement of antioxidant defense along with reduced glutathione level, which ultimately reduces the cellular oxidative stress.

## 1. Introduction

α-Lipoic acid (ALA), a dithiol compound, which is often reduced enzymatically to dihydrolipoic acid (DHLA), is well known for its significant antioxidant properties. ALA and DHLA have gained interest because of their potential role in (a) free-radical scavenging, (b) metal chelation, and (c) restoring intracellular glutathione (GSH) levels against environmental pollutants, such as heavy metals [1,2,3]. Recently, ALA has been used in multivitamin formulations, food supplements, antiaging formulas, and even human and pet food recipes, as an additional antioxidant compound [1,4]. The structures of ALA and DHLA are depicted in Figure 1.

In recent years, ALA has been studied extensively as a biological antioxidant, detoxification agent, and antidiabetic medicine [1,5]. Moreover, ALA has been claimed to ameliorate age-associated cardiovascular, cognitive, and neuromuscular deficits; it has also been implicated as a modulator of various inflammatory signaling pathways [6,7,8,9,10].

In addition, environmental chemicals have received increasing attention, and they have become a major concern in the field of oxidative stress and disease-promoting actions due to their frequent exposure to food chains, general air, and water pollution. In particular, the toxic metals arsenic (As), cadmium (Cd), and lead (Pb) represent hazardous environmental contaminants and are responsible for numerous diseases. It is crucial to investigate better environmental agents/solutions to eliminate the toxicity of metals. ALA and DHLA with their antioxidant properties can be potential therapeutic mediators against metal toxicity. In this study, we hypothesized that ALA and DHLA have cytoprotective effects against metal-induced toxicity in cells. To examine the possible role of ALA and DHLA against these toxic metals, we conducted an in vitro study using mammalian cells (PC12 and Caco-2). PC12 cells (neuron-like rat brain cells), which are well known and commonly used as a model cell line for toxicity assessment [11,12,13], were used in this research. Moreover, to evaluate the effects of ALA and DHLA in a cancerous pathological situation, Caco-2 cells (human colorectal cancer cells) were employed.

There are no previous reports comparing ALA protection against As-, Cd-, and Pb-induced toxicity in two different cell lines under the same treatment conditions. For the first time, the effect of ALA on As-, Cd-, and Pb-induced toxicity was evaluated in PC12 and Caco-2 cells. Moreover, there are no available studies on the amelioration of DHLA against As-, Cd-, and Pb-induced toxicity in PC12 cells, which was investigated in this study. In addition, our study provides a hint of comparison on the protection ability of ALA and DHLA in the PC12 cell line in terms of antioxidant properties. Furthermore, the cytoprotective potential was also evaluated by focusing on the mechanisms of action. Therefore, to address the abovementioned hypothesis, the protective effects of ALA/DHLA were examined in metal-induced toxicity using various cytotoxicity assessment methods.

## 2. Materials and Methods

### 2.1. Materials

PC12 cells were purchased from the American Type Culture Collection (USA and Canada) and Caco-2 cells (RCB0988) were kindly donated by the RIKEN BRC through the National Bio-Resource Project of the MEXT, Japan. Dulbecco’s modified Eagle’s medium (DMEM), ribonuclease A (RNase), ethidium bromide, and peroxidase-conjugated avidin were purchased from Sigma (St. Louis, MO, USA). Eagle’s minimum essential medium (EMEM) was purchased from Wako (Japan). Fetal bovine serum (FBS) was obtained from Biosera (Kansas City, MO, USA). The SV total RNA isolation system and RT-PCR kit were purchased from Promega (Madison, WI, USA). The high pure PCR product purification kit and proteinase K were purchased from Roche Diagnostics (Mannheim, Germany). Biotinylated goat anti-mouse IgG (immunoglobulin G) whole antibody and enhanced chemiluminescence (ECL) Western blotting detection reagent were purchased from Amersham Pharmacia Biotech (Buckinghamshire, England). Polyclonal antibodies against specific proteins such as Nrf2 (nuclear factor erythroid 2-related factor 2) (PM069, Medical and Biological Laboratories Co., Japan), mTOR (mammalian target of rapamycin) (2972, Cell Signaling Technology), Akt (protein kinase B) (4691, Cell Signaling Technology), cleaved PARP-1 (poly(ADP-ribose) polymerase-1) (ab32064, Abcam), and β-actin (GTX 109639, GeneTEX) were purchased. Trypan blue solution (0.4%) was purchased from Bio-Rad (Hercules, CA, USA). The DNA 7500 assay and RNA 6000 nano assay kits were purchased from Agilent Technologies (Waldbronn, Germany). All other chemicals were of analytical grade.

### 2.2. Cell Culture

PC12 cells were cultured in DMEM supplemented with 10% FBS in a humidified incubator at 37°C with 5% CO_2_, whereas Caco-2 cells were cultured in EMEM medium supplemented with 10% FBS and 1% nonessential amino acids (NEAA) under the same incubation conditions. The cells were preincubated in 25 cm^2^ flasks for 24 h; then, the medium was replaced with fresh medium with or without various concentrations of Cd, As, Pb, ALA, DHLA, or with a mixture of them, and the cells were re-incubated for another 48 h. The desired concentration for treatment was selected by exposing PC12 and Caco-2 cells to several concentrations of each agent; for example, cells were exposed to 0, 2, 5, 10, and 20 µM As^3+^ separately, and then the final combination was decided. The selected concentrations for As^3+^, Cd^2+^, Pb^2+^, ALA, and DHLA were 5, 5, 5, 250, and 50 µM, respectively. The effects of ALA and DHLA were examined in PC12 and Caco-2 cells.

### 2.3. Cell Viability

Cell viability was determined using the trypan blue exclusion assay. PC12 and Caco-2 cells were seeded at a density of 1 × 10^5^ cells/flask and preincubated for 24 h. Then, the cells were treated with As^3+^, Cd^2+^, Pb^2+^, ALA, and DHLA separately, as well as co-exposed to As^3+^ (5 μM), Cd^2+^ (5 μM), Pb^2+^ (5 μM), ALA (250 μM), and DHLA (50 μM). The cells were then incubated for 48 h. After 48 h, cells were harvested, washed using 1× phosphate-buffered saline (PBS), and incubated with trypan blue dye. To measure the stained and non-stained cells, an automated cell counter (Hercules, CA, USA) was employed, in which non-stained cells indicate live/viable cells. Each experiment was performed at least in triplicate to ensure biological reproducibility and statistical validity.

### 2.4. Lactate Dehydrogenase (LDH) Activity Assay

Cytotoxicity was assessed by measuring the activity of LDH in the treatment medium using a nonradioactive cytotoxicity assay kit (Promega), as described by Kihara et al. [14]. PC12 and Caco-2 cells (1 × 10^5^ cells/flask) were cultured in medium with or without As^3+^ (5 μM), Cd^2+^ (5 μM), Pb^2+^ (5 μM), ALA (250 μM), and DHLA (50 μM) for 48 h. After 48 h of incubation, 50 µL of the medium was transferred to a 96-well plate, and then 50 µL of the substrate mixture containing tetrazolium salts was added to each of the sample wells in a 96-well plate. After a 30 min incubation at room temperature (25°C), 50 µL of the stop solution was added, and the amount of formazan dye formed was determined by measuring the absorbance at 490 nm using a microplate reader (Bio-Rad, CA, USA). LDH activity was expressed as LDH activity/1 × 10^6^ cells. This experiment was carried out in triplicate to ensure reproducibility.

### 2.5. Measurement of GSH Levels

Intracellular glutathione (GSH) levels were investigated as previously described by Kihara et al. [14]. Cells (1 × 10^5^) were preincubated for 24 h and subsequently exposed to As^3+^ (5 μM), Cd^2+^ (5 μM), Pb^2+^ (5 μM), ALA (250 μM), and DHLA (50 μM) for 48 h. The cells were harvested, washed with 1× phosphate-buffered saline (PBS), added to 150 µL of lysis buffer, and then incubated at room temperature (25 °C) for 10 min. Two freeze–thaw sonication cycles were performed in order to rupture the cell membranes, and the resultant solution was centrifuged at 1500 rpm for 10 min to collect the supernatant. Intracellular free GSH levels were determined using 2.5 μmol/L 5,5′-dithiobis-2-nitrobenzoic acid (DTNB, pH 7). DTNB (final concentration; 20 μM) was added to the cell lysate, and the absorbance was measured at 412 nm using a microplate reader (Bio-Rad, CA, USA). The concentration of intracellular GSH in cells was determined using a molecular coefficient factor of 13,600 per cell number (1 × 10^5^ cells). The experiment was performed in triplicate to ensure reproducibility.

### 2.6. Isolation of Genomic DNA and Agarose Gel Electrophoresis

After treatment of PC12 and Caco-2 cells with the abovementioned chemicals, the cells were harvested. Then, the obtained cells were centrifuged at 1500 rpm for 5 min to remove the supernatant. After the addition of 3 mL of PBS, the mixture was centrifuged again at 1500 rpm for 5 min to wash the cells. Genomic DNA was isolated using the high pure PCR template preparation kit according to the manufacturer’s instructions, as described by Hossain et al. [15]. The obtained solution containing DNA was mixed with 2 μL of 500 μg/mL RNase and incubated for 15 min at 37 °C. After incubation, 500 μL of ethanol and 20 μL of a 3 M NaOAc buffer (pH 4.5) were added for ethanol precipitation, and the solution was allowed to stand overnight in a freezer to precipitate the DNA. The next day, the DNA was separated by using microcentrifugation at 15,000 rpm for 8 min, and then washed with 70% ethanol at the same speed for 3 min. Then, the DNA sample was dried for approximately 10 min, and DNA concentration was measured after the reaction with 1× TBE (Tris/Borate/**** Ethylenediamine tetraacetic acid) using an ultraviolet (UV)–visible light spectrophotometer. The ladder pattern/intact DNA was analyzed via agarose gel electrophoresis. Approximately 3–5 μg of DNA with the loading dye was subjected to electrophoresis on a 1.5% agarose gel. Electrophoresis was carried out for 40 min at 100 V in a 1.5% of agarose gel using a submarine-type electrophoresis system (Mupid-ex, Advance, Tokyo, Japan). To visualize the DNA fragmentation/degradation, the gel was soaked in an ethidium bromide solution for 5–10 min. Images of the agarose gel were taken under UV illumination using a ChemiDoc XRS (Bio-Rad, Hercules, CA, USA). To evaluate cell apoptosis/DNA damage, the fluorescence intensity of DNA in the gel was analyzed using a software named Quantity one. The density of the DNA band was analyzed using ImageJ software. This experiment was conducted at least in triplicate.

### 2.7. Western Blot Analysis for Determination of Protein Expression

Cells were cultured in 5 mL of DMEM/EMEM containing 10% FBS, with or without the abovementioned treatment for 48 h. The Western blot analysis procedure was performed according to Rahman et al. [16]. Briefly, the total protein in cells was extracted using ice-cold lysis buffer (2mM HEPES (4-(2-hydroxyethyl)-1-piperazineethanesulfonic acid), 100 mM NaCl, 10 mM EGTA (Ethylene glycol tetraacetic acid), 0.1 μM PMSF (phenylmethanesulfonyl fluoride), 1 mM Na_3_VO_4_, 0.1 mM Na_2_MgO_4_, 5 mM 2-glycerophosphoric acid, 10 μM MgCl_2_, 2 mM DTT (dithiothreitol), 50 μM NaF, and 1% Triton X-100). The extracted protein concentration was measured spectrophotometrically using a protein assay dye reagent (BioRad). The total protein (30 µg) from each sample was separated using 12.5%/15% sodium dodecyl sulfate-polyacrylamide (SDS-PAGE) electrophoresis and then transferred onto nitrocellulose membranes using the semi-dry transfer method. After that, the membrane was blocked with 5% skimmed milk at 4 °C for 24 h. Next, the membrane was sequentially incubated with the desired primary and secondary antibodies. Finally, the protein on the nitrocellulose membrane was visualized using enhanced chemiluminescence and the image of the detected band was analyzed using ChemiDoc XRS (Bio-Rad, USA). The density of the protein bands was analyzed using ImageJ software. The intensities of the bands were compared to that of β-actin (internal control). All experiments were conducted at least in triplicate to ensure reproducibility.

### 2.8. Statistical Analysis

All data are expressed as the mean ± standard error of the mean (SEM). Statistical analyses were performed using single-factor analysis of variance (ANOVA) followed by unpaired Student’s *t*-test.

## 3. Results

### 3.1. Combined Effects of Toxic Metals and ALA/DHLA on Cell Viability

PC12 cells were treated with As (0–20 μM) (Figure 2A), Cd (0–20 μM) (Figure 2B), and Pd (0–20 μM) (Figure 2C) for 48 h, and a concentration of 5 μM was observed to be close to lethal dose (LD_50_) values (4.83, 4.46, and 4.37 μM, respectively). From these observations, a 5 μM concentration of the metals was selected for further experiments. The viability of PC12 and Caco-2 cells after exposure to ALA (125–1000 μM) for 48 h was determined using the trypan blue exclusion method (Figure 2D). It was observed that an ALA concentration up to 250 μM did not have any harmful effect on cells. However, an ALA concentration higher than 250 μM caused a significant decrease in cell viability of Caco-2 cells (Figure 2D). Therefore, concentrations up to 250 μM (125 and 250 μM) were observed to protect cells from metal-induced toxicity (Figure 2E–G). ALA at a concentration of 250 μM was found to provide a better protection than 125 μM ALA. For this reason, only 250 μM ALA was selected to be used in the combined study with toxic metals (As, Cd, or Pb).

In addition, cells were exposed to a suitable range (0–150 μM) of DHLA (Figure 2H). Up to 150 μM of DHLA was found to have no toxic effect on cells. The lowest concentration (50 μM) was chosen for the combined metal experiment.

To examine the combined effect of As, Cd, or Pb with ALA and DHLA cells were exposed to both chemicals (binary mixture) for 48 h (Figure 3 and Figure 4). Results showed that, in both PC12 and Caco-2 cells, As (5 μM), Cd (5 μM), and Pb (5 μM) induced significant cell death compared to the control group (Figure 3). However, in co-exposure with ALA (250 μM) and toxic metals, cell viability increased significantly compared to the metal-only group for both PC12 and Caco-2 cells (Figure 3A,B).

In addition, DHLA (50 μM) showed cytoprotection by significantly increasing cell viability in combination with As (5 μM), Cd (5 μM), and Pb (5 μM) in PC12 cells (Figure 4). A remarkable increase in cell viability was observed after the combined treatment with DHLA and toxic metals (Figure 4), in which there was no significant difference between the control group and the combined treatment (metals + 50 μMDHLA) group.

### 3.2. Combined Effects of Toxic Metals and ALA/DHLA on Cell Membrane Integrity

LDH activity was measured in the cell culture medium after being exposed to As, Cd, Pb, ALA, and DHLA for 48 h. Both PC12 and Caco-2 cells showed a significant increase in LDH activity in their respective culture medium in response to metal burdens, indicating a loss of cell membrane integrity (Figure 5 and Figure 6). However, the co-exposure to toxic metals and ALA/DHLA resulted in a significant decrease in LDH activity compared to the metal-treated group alone in PC12 and Caco-2 cells (Figure 5 and Figure 6). These results corroborate the finding of cell viability under similar experimental conditions for both cell lines. The cell membrane damage was significantly lower in the co-treated group (toxic metals + ALA/DHLA) than in the metal-treated group alone.

### 3.3. Combined Effects of Toxic Metals and ALA/DHLA on GSH Level

Intracellular GSH levels were measured following metal exposure with/without ALA/DHLA in PC12 and Caco-2 cells. As, Cd, and Pb exposure significantly decreased the GSH levels in both cell lines (Figure 7 and Figure 8A). ALA and DHLA both showed a significant inhibition of GSH oxidation due to As, Cd, or Pb exposure upon co-exposure in PC12 cells. Again, ALA showed a similar protection effect against metal-induced GSH depletion in Caco-2 cells (Figure 7B). A severe depletion of GSH was observed upon exposure to toxic metals in both PC12 and Caco-2 cells. Therefore, there was a possible increase in oxidative stress due to the burden of toxic metals in cells. However, exposure to ALA and DHLA was found to boost the GSH defense against toxic metal stress in both PC12 and Caco-2 cells.

### 3.4. Combined Effects of Toxic Metals and ALA/DHLA on DNA Damage

To further investigate DHLA protection against DNA damage or DNA fragmentation, agarose gel electrophoresis was employed. The results of agarose gel electrophoresis revealed that the cellular intact DNA was decreased significantly in PC12 cells due to metal exposure, when compared to the control group, thereby indicating severe DNA fragmentation/damage (Figure 8B,C). Interestingly, the co-exposure of cells to metals and DHLA showed a significant increase in intact DNA density compared to the metal-treated group alone, thereby indicating a protection effect induced by DHLA. DNA fragmentation caused by As and Cd was also characterized by ladder/smearing pattern in PC12 cells, which indicates the possibility of apoptosis, a type of cell death. On the other hand, in the case of Pb, there was no ladder/smearing pattern in PC12 cells, which could not confirm apoptosis. These results indicated that the metals (in the separate exposure) induced severe DNA damage, and DNA damage was significantly inhibited upon co-exposure with DHLA.

### 3.5. Combined Effects of Toxic Metals and ALA/DHLA on Protein Expressions

The molecular mechanisms responsible for ALA/DHLA-induced defense against metal-influenced cytotoxicity were investigated using Western blotting analysis (Figure 9, Figure 10 and Figure 11) after co-treatment with As, Cd, and Pb.

In PC12 cells, metals downregulated the expression of mTOR and Akt proteins, which are responsible for cellular survival, cell growth, metabolism, differentiation, and cell-cycle progression, compared to the control (Figure 9). Moreover, upon metal exposure, the expression of Nrf2, which is responsible for regulating expressions of antioxidant proteins, was downregulated compared to the control group (Figure 9). However, the co-exposure using ALA and metals (As and Cd) upregulated the expression of Nrf2, compared to the group exposed to metals alone (Figure 9). Furthermore, ALA upregulated the expression of mTOR and Akt, which were downregulated by the exposure to As (5 μM), Cd (5 μM), and Pb (5 μM) in PC12 cells (Figure 9).

The protein expression levels after treatment with metals and ALA in Caco-2 cells are shown in Figure 10. From the immunoblot image, it was observed that the metals induced an upregulation of the proapoptotic cleaved PARP-1 (poly(ADP-ribose) polymerase-1; a hallmark of apoptotic cell death) and a downregulation of Nrf2 protein expression. These findings suggested that the metals induced an apoptotic cell death, which supports the DNA analysis results obtained (Figure 8B). On the other hand, ALA co-exposure significantly reversed the protein expression of cleaved PARP-1 and Nrf2 in the case of As and Cd (Figure 10), whereas ALA co-exposure with Pb showed a reverse trend, although not significant, when compared to the metal-treated group alone.

In PC12 cells, DHLA (50 μM) co-exposure upregulated the expression of pro-survival mTOR, Akt, and Nrf2 proteins, which were downregulated by exposure to As (5 μM), Cd (5 μM), or Pb (5 μM) in PC12 cells (Figure 11).

## 4. Discussion

ALA has antioxidant potential under oxidative stress conditions. In the present study, we found an interesting protective effect of ALA and DHLA on As-, Cd-, and Pb- induced toxicity in PC12 and Caco-2 cells, whereas ALA/DHLA alone showed no significant toxic effect on both PC12 and Caco-2 cells. ALA and DHLA detoxify As-, Cd-, and Pb-induced toxicity by limiting LDH activity and decreasing the GSH content, thereby reducing the oxidative stress. Moreover, ALA and DHLA detoxify cells from As- and Cd-induced toxicity by activating Nrf2, influencing mTOR, inducing Akt upregulation, inhibiting DNA fragmentation, and protecting against cell death.

As ALA showed functional pleiotropism via different signal transduction pathways, its use as a potential therapeutic agent is quite promising [3,5,10]. ALA and DHLA can scavenge numerous free radicals such as reactive oxygen and nitrogen species [17,18]. Furthermore, DHLA maintains GSH and protein thiols in their reduced forms. Dietary intake of ALA has been considered to be clinically safe, because ALA enters biological membranes easily due to its affinity with membrane lipids, as well as the blood–brain barrier, conferring it a potential neurodegenerative preventive activity [19,20,21]. Unlike other antioxidants, after reacting with various prooxidants and free radicals, ALA can easily reconstitute DHLA. Furthermore, ALA and DHLA are amphipathic molecules and can, therefore, work both in aqueous and in hydrophobic environments, which enables them to counteract the oxidation of lipids, proteins, and DNA [22,23].

In the present study, ALA and DHLA alone did not induce oxidative stress in PC12 and Caco-2 cells compared to the control group. These results are consistent with the findings of a study reporting that ALA (200 μM) did not affect colorectal cancer cells [24]. As, Cd, and Pd can induce oxidative stress via intracellular reactive oxygen species (ROS) generation and the oxidation of reduced GSH, which could challenge the homeostasis of cellular macromolecules such as proteins, nucleic acids, and lipids [15,16]. In our study, a significant reduction in cell viability and a loss of cell membrane integrity were observed upon exposure to As, Cd, and Pb in PC12 and Caco-2 cells (Figure 1 and Figure 2). As and Cd induced a similar reduction in cell viability and cell membrane damage in PC12 cells to that reported in previous studies [11,16]. Interestingly, the simultaneous exposure to ALA/DHLA significantly decreased metal-induced cell viability loss and cell membrane damage in both PC12 and Caco-2 cells (Figure 3 and Figure 4), which is a clear indication of the cytotoxicity inhibition in the presence of ALA and DHLA.

Intracellular GSH levels were significantly decreased upon exposure to As, Cd, or Pb in PC12 and Caco-2 cells, indicating an induction of oxidative stress. For instance, Cd may directly form conjugates with thiol groups in GSH to induce a GSH–Cd conjugate, which can readily be excreted out of the cells, thus depleting the reduced GSH [25]. Lawal and Ellis [26] reported that an excessive ROS production, along with the oxidation of reduced GSH, might be crucial in Cd-induced cytotoxicity in three human cell lines: HepG2 cells, 1321N1 cells, and HEK293 cells. Similarly, As induced the depletion of GSH in PC12 cells [16], similar to Cd. Therefore, in this study, the metal-induced cytotoxicity could be at least partially related to the lowering of the intracellular GSH, which may disrupt the GSH/glutathione disulfide (GSSG) ratio. However, the co-exposure to these metals with ALA and DHLA significantly increased the level of GSH in the studied cell lines (Figure 5 and Figure 6). A similar increase in the GSH levels upon DHLA exposure against metal-induced toxicity has been reported [27]. Therefore, metal-induced oxidative stress can be reduced by the co-exposure with ALA/DHLA. Increased levels of GSH in co-exposure with metals may be the key mechanism of ALA/DHLA to resist metal-induced GSH oxidation, thus protecting the cells (Figure 5 and Figure 6). Our present findings corroborate the findings of Bharat et al. [28], showing that ALA protected dopaminergic PC12 cells from oxidative stress and cytotoxicity by preventing the depletion of GSH content.

Nrf2 is a transcription factor activated in response to oxidative stress [29,30,31]. In our study, the As-, Cd-, and Pb-treated groups showed a downregulation of Nrf2 protein expression. Interestingly, the co-exposure to ALA and DHLA with metals showed an increase in Nrf2 protein level (Figure 7 and Figure 8), indicating that ALA and DHLA both induced antioxidant (ROS scavenging) events. Our findings are corroborated by a study conducted in HepG2 cells by Shi et al. [32], in which they found a downregulation of Nrf2 upon Cd exposure, and this downregulation was reversed by the co-exposure of HepG2 cells to Cd and ALA [32,33]. However, Nrf2 upregulation by ALA against Pb-induced toxicity was not significant upon co-exposure in PC12 and Caco-2 cells, but it was significant when co-exposed with DHLA; therefore, further investigation with Pb is needed. Moreover, in our study, the downregulated expression of mTOR and Akt upon As, Cd, and Pb exposure in PC12 cells might be characterized by cellular stress and cytotoxicity. A similar downregulation of mTOR and Akt was observed upon exposure to metals in PC12 cells and colorectal cancer cells (HCT116 and Caco-2 cells), indicating cellular stress [16,24]. Furthermore, the co-exposure to ALA/DHLA reversed the situation in PC12 cells. However, metals induced cleavage of PARP-1 in Caco-2 cells, which, together with the DNA damage results, is a direct indication of apoptosis. This suggested that metals induced cell death via the apoptosis process, and the co-exposure to ALA with As and Cd subsequently reduced the expression of cleaved PARP-1. A similar cleaved PARP-1 induction was reported in Caco-2 cells, which explains the apoptosis [24]. The normal function of PARP-1 is the routine repair of DNA damage [34]. Cleavage of PARP-1 by caspases is considered a hallmark of apoptosis [35]. In addition, in PC12 cells, severe DNA damage was induced by As, Cd, or Pb, which was reduced upon co-exposure to DHLA with metals (Figure 6B,C). A similar DNA fragmentation induced by Cd or As was reported previously in PC12 cells [15,16].

In summary, As, Cd, and Pb were found to increase the cellular oxidative stress and the subsequent alterations in biomolecules, such as proteins, DNA, and GSH in both PC12 and Caco-2 cells, and significant ameliorative effects were observed following the treatment with ALA/DHLA. Thus, ALA and DHLA can detoxify metal toxicity in PC12 and Caco-2 cells by reducing oxidative stress, activating Nrf2, inhibiting DNA fragmentation, reducing cell membrane damage, and protecting against cell death (Figure 6B,C). Our results suggested some differences in the cytoprotective molecular mechanisms induced by ALA in different cell lines, but also supported similar protective mechanisms induced by ALA and DHLA in the same cell line. It was observed that, irrespective of the cell lines (PC12 or caco-2), the Nrf2 pathway was involved in both ALA and DHLA protection, with some exceptions regarding Pb toxicity. Further research is needed to better understand the underlying molecular mechanisms. Lastly, the GSH level increase, along with activation of antioxidant pathways, seems to be the key protective mechanism provided by ALA and DHLA.

## 5. Conclusions

In this study, significant cytoprotective effects of ALA and DHLA were reported against As-, Cd-, and Pb-induced toxicity in PC12 and Caco-2 cells. The strengthening of the cellular antioxidant defense by ALA and DHLA could be a crucial way of eliminating the metal-induced cellular burden in cells. However, the regeneration of a reduced GSH level, along with the Nrf2 signaling pathway, might be involved in the antioxidant defense regime. It can be concluded that ALA/DHLA has protective roles against toxic metal-induced cellular damage in both PC12 and Caco-2 cell lines.

## Figures and Tables

**Figure 1 ijerph-18-02126-f001:**
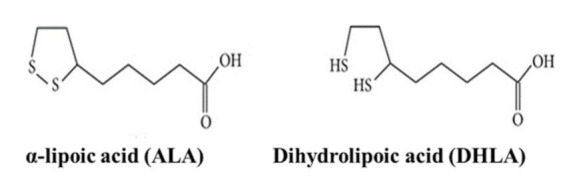
Structures of α-lipoic acid (ALA) and dihydrolipoic acid (DHLA).

**Figure 2 ijerph-18-02126-f002:**
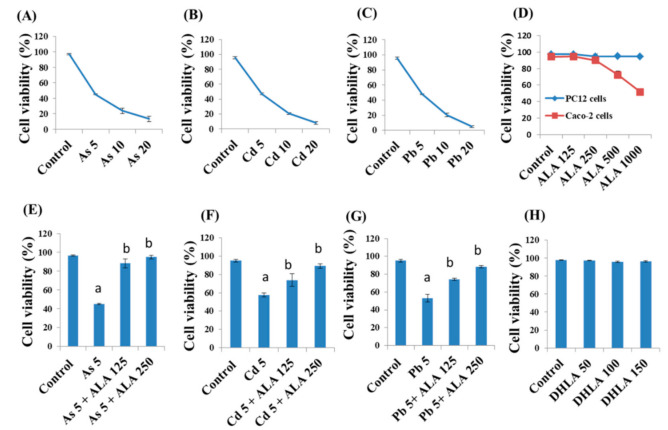
Cell viability of PC12 cells after exposure to toxic metals (**A**) As (0–20 μM), (**B**) Cd (0–20 μM), and (**C**) Pb (0–20 μM) for 48 h of incubation. Cell viability of PC12 cells and Caco-2 cells after exposure to (**D**) ALA (0–1000 μM) for 48 h of incubation. Cell viability of PC12 cells after exposure to (**E**) As (5 μM) (**F**) Cd (5 μM), and (**G**) Pb (5 μM) and/or ALA (125 and 250 μM) for 48 h of incubation. (**H**) Cell viability of PC12 cells after exposure to DHLA (0–150 μM) for 48 h of incubation. The valued of the bars indicate the mean ± standard error of the mean (SEM) (*n* = at least 3). Here, “a” denotes a significant difference compared to the control group (*p* < 0.05), and “b” denotes a significant difference compared to the associated metal group (*p* < 0.05).

**Figure 3 ijerph-18-02126-f003:**
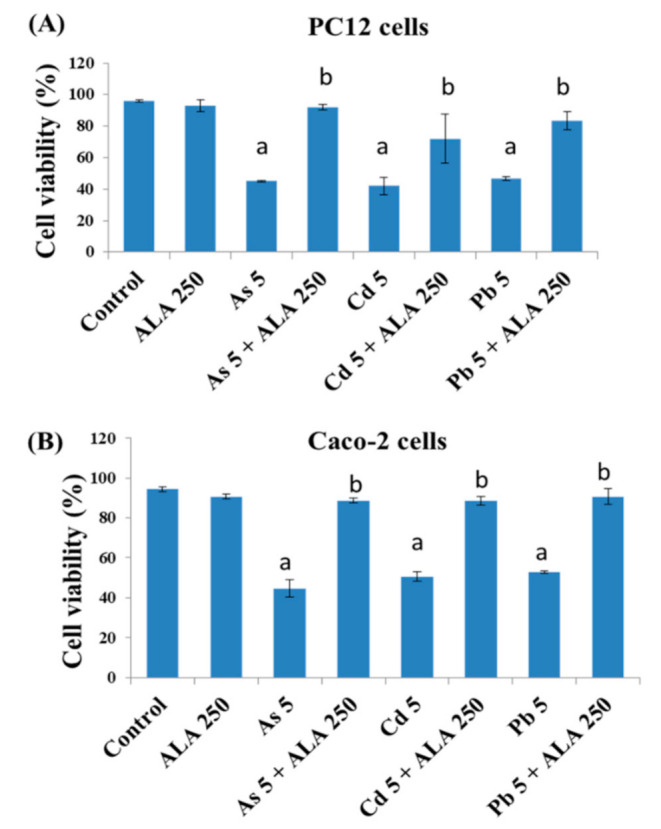
Cell viability of (**A**) PC12 cells and (**B**) Caco-2 cells after exposure to toxic metals and/or ALA for 48 h of incubation. The values of the bars indicate the mean ± SEM (*n* = 5). Here, “a” denotes a significant difference compared to the control group (*p* < 0.05), and “b” denotes a significant difference compared to the associated metal group (*p* < 0.05).

**Figure 4 ijerph-18-02126-f004:**
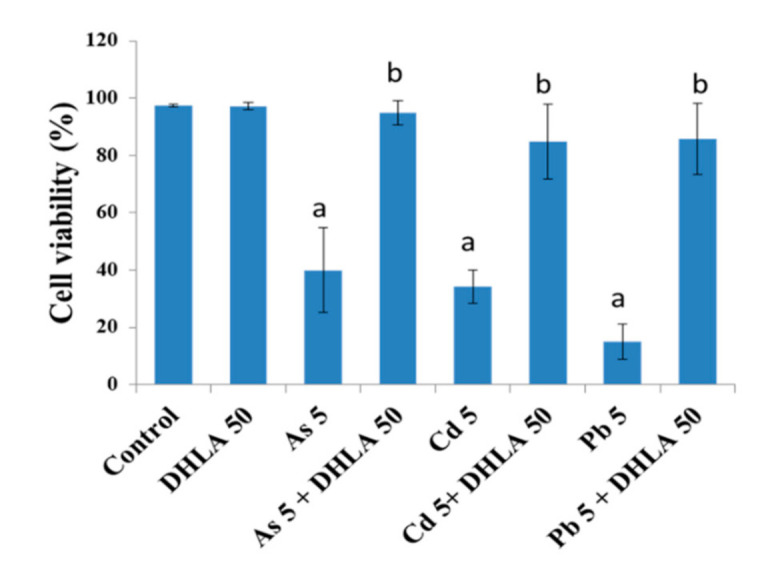
Cell viability of PC12 cells after treatment with toxic metals and/or DHLA for 48 h determined by trypan blue exclusion method. The values of the bars indicate the mean ± SEM (*n* = 5). Here, “a” denotes a significant difference compared to the control group (*p* < 0.05), and “b” denotes a significant difference compared to the associated metal group (*p* < 0.05).

**Figure 5 ijerph-18-02126-f005:**
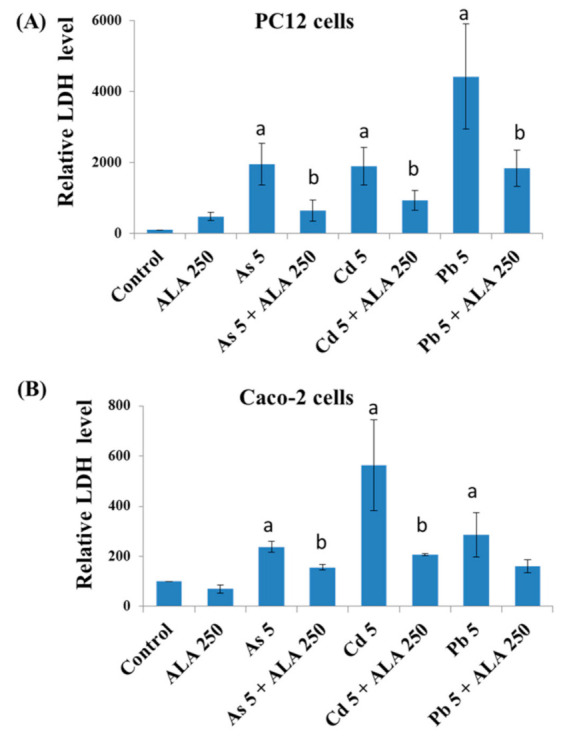
Lactate dehydrogenase (LDH) activity of (**A**) PC12 cells and (**B**) Caco-2 cells in cell culture medium after exposure to toxic metals and/or ALA for 48 h of exposure. The values of the bars indicate the mean ± SEM (*n* = 3). Here, “a” denotes a significant difference compared to the control group (*p* < 0.05), and “b” denotes a significant difference compared to the associated metal group (*p* < 0.05).

**Figure 6 ijerph-18-02126-f006:**
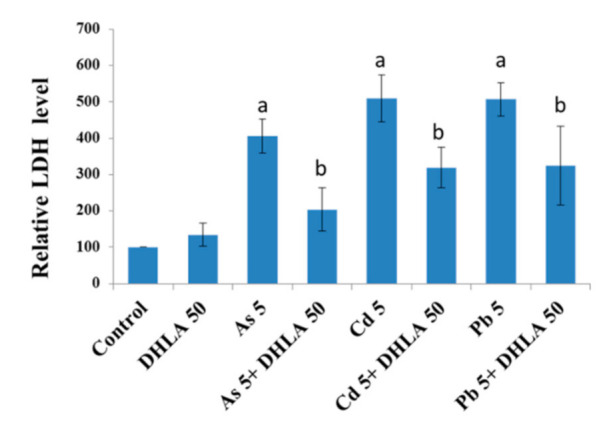
Lactate dehydrogenase (LDH) activity in cell culture medium after exposure to toxic metals and/or DHLA for 48 h of exposure. The values of the bars indicate the mean ± SEM (*n* = 3). Here, “a” denotes a significant difference compared to the control group (*p* < 0.05), and “b” denotes a significant difference compared to the associated metal group (*p* < 0.05).

**Figure 7 ijerph-18-02126-f007:**
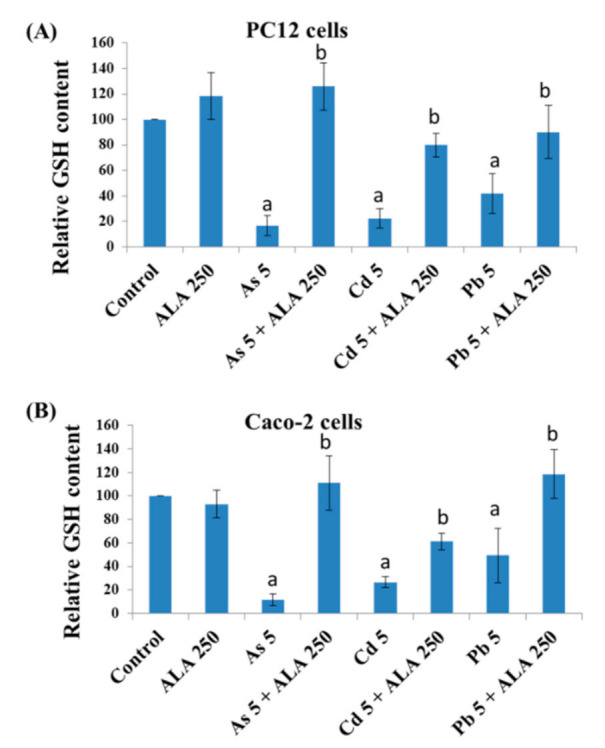
Intracellular levels of glutathione (GSH) upon exposure to toxic metals and/or ALA measured by the 5,5′-dithiobis-2-nitrobenzoic acid (DTNB) assay in (**A**) PC12 cells and (**B**) Caco-2 cells. The values of the bars indicate the mean ± SEM (*n* = 3). Here, “a” denotes a significant difference compared to the control group (*p* < 0.05), and “b” denotes a significant difference compared to the associated metal group (*p* < 0.05).

**Figure 8 ijerph-18-02126-f008:**
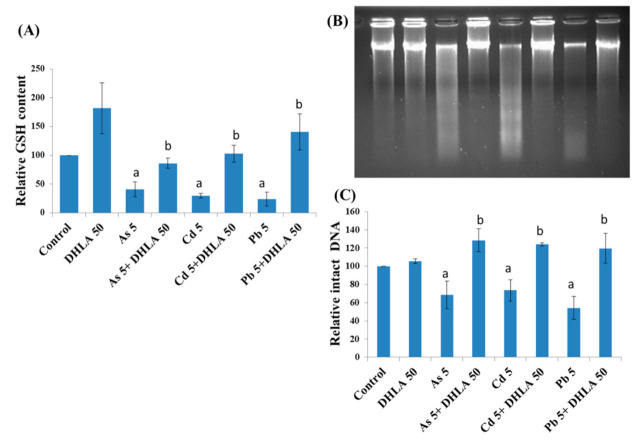
(**A**) Intracellular levels of GSH upon exposure to toxic metals and/or DHLA measured by the DTNB assay in PC12 cells. (**B**) DNA electrophoresis taken by agarose gel method and (**C**) DNA density measurement calculated by image J software after co-treatment with metals and DHLA in PC12 cells. The values of the bars indicate the mean ± SEM (*n* = 3 and *n* = 4, respectively). Here, “a” denotes a significant difference compared to the control group (*p* < 0.05), and “b” denotes a significant difference compared to the associated metal group (*p* < 0.05).

**Figure 9 ijerph-18-02126-f009:**
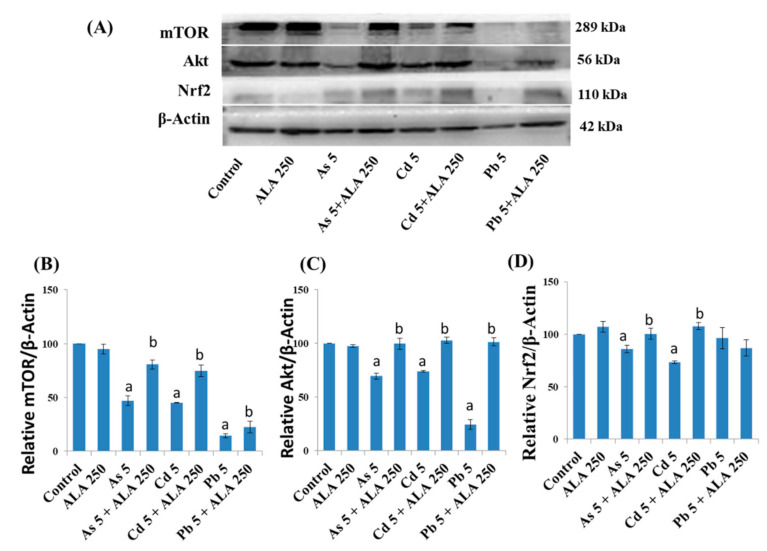
(**A**) Western blot analysis of protein expressions, and quantification of (**B**) mTOR, (**C**) Akt and (**D**) Nrf2 in the PC12 cells after treatment with metals with ALA for 48 h. The values of the bars indicate the mean ± SEM (*n* = 3). Here, “a” denotes a significant difference compared to control group (*p* < 0.05), and “b” denotes a significant difference compared to the associated metal group (*p* < 0.05).

**Figure 10 ijerph-18-02126-f010:**
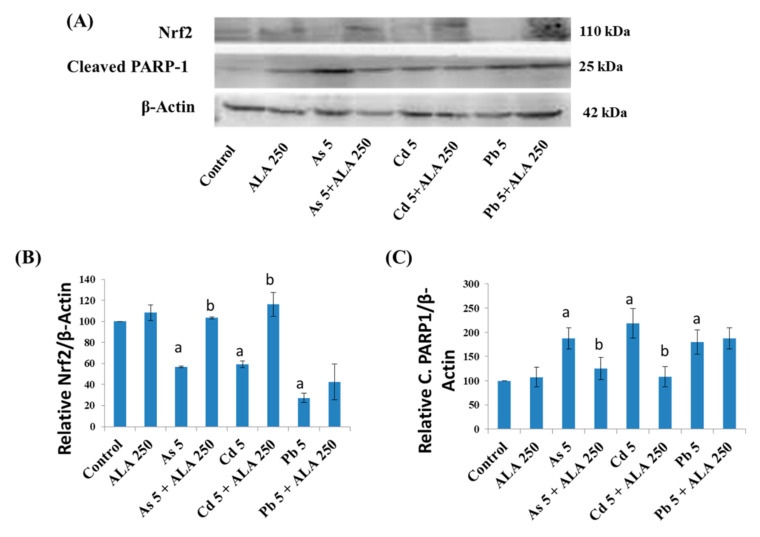
(**A**) Western blot analysis of protein expressions, and quantification of (**B**) Nrf2 and (**C**) cleaved PARP1 in the Caco-2 cells after treatment with metals with ALA for 48 h. The values of the bars indicate the mean ± SEM (*n* = 3). Here, “a” denotes a significant difference compared to the control group (*p* < 0.05), and “b” denotes a significant difference compared to the associated metal group (*p* < 0.05).

**Figure 11 ijerph-18-02126-f011:**
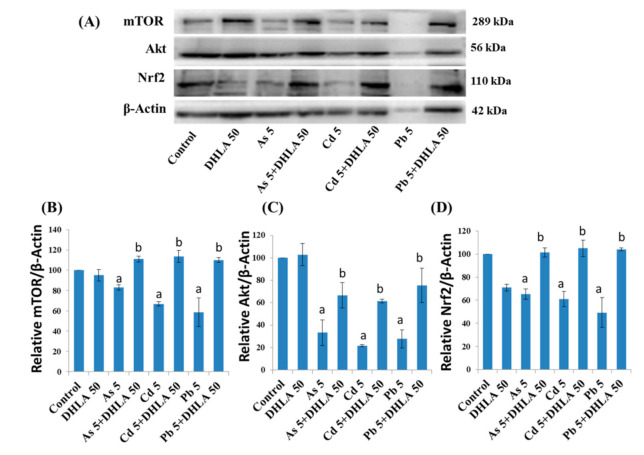
(**A**) Western blot analysis of protein expressions, and quantification of (**B**) mTOR, (**C**) Akt and (**D**) Nrf2 in the PC12 cells after treatment with metals with DHLA for 48 h. The values of the bars indicate the mean ± SEM (*n* = 3). Here, “a” denotes a significant difference compared to the control group (*p* < 0.05), and “b” denotes a significant difference compared to the associated metal group (*p* < 0.05).

## Data Availability

All data is presented in this study and thus contained within the article. There are no other available data in any publicly accessible repository.

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
