# Peer review of "Amelioration of Metal-Induced Cellular Stress by α-Lipoic Acid and Dihydrolipoic Acid through Antioxidative Effects in PC12 Cells and Caco-2 Cells"

_ijerph, 2021, doi:10.3390/ijerph18042126_

Round 1
Reviewer 1 Report
The current research aims to investigate where Alpha Lipoic Acid (LA) and Dihydrolipoic Acid (DHLA) with antioxidant properties c have cytoprotective effects against arsenic (As), cadmium (Cd) and lead (Pb)-induced metal toxicity with the cell model of mammalian cells PC12 and Caco-2. Based on the results from cell viability assays, LDH activity assays, intracellular GSH levels, genomic DNA integrity, and Western blotting analyses, the authors suggested that LA and DHLA are capable of reducing the toxic effects of metals, possibly due to enhancement of the antioxidant defense by reducing cellular oxidative stress.
However, there were some critical concerns about the research design and manuscript writing. Moreover, the results at the current status did not support the hypothesis.
- Basic information of cell line PC12 and Caco-2 should be described. Moreover, the rationale for the use of these two cell models should be described. What is the pathophysiological significance of using these two cell models?
- The rationale for the concentration of LA (250 μM) and DHLA (50 μM) should be described. The simple description of data not shown (Line 184-185) did not provide useful information on whether LA (250 μM) and DHLA (50 μM) were the optimal doses. Why did not use the same concentration of LA and DHLA, as Dihydrolipoic Acid (DHLA) is the reduced form of Alpha Lipoic Acid (LA)?
- The rationale for the concentration of Cd (5 μM), As (5 μM), and Pb (5 μM)? Are these concentrations related to their plasma concentration in patients with metal toxicity?
- Figures 7 & 8: The control, ALA/DHLA, metal, and metal+ALA/DHLA should be detected in the same gel/membrane. It is impossible to compare the protein expression levels of two groups from two or more different blots!
- Structures of Alpha Lipoic Acid (LA) and Dihydrolipoic Acid (DHLA) should be provided in the manuscript.
- Materials & Methods: (Line 112) Cell viability was expressed as the percentage of the counted trypan blue-stained cells. As the cell viability represented the percentage of viable cells, the way to express cell viability should be another way around.
- Statistical analyses are missing in the section of Materials & Methods!
- Figures: what did “a, b, c” on the bar represent?
- Figure legends: The “mean ± SEM” should be the value of bars, but not “error bars.”
Reviewer 2 Report
The manuscript provided an evaluation of the reduction of the metal toxic effects by alpha lipoic acid (ALA/LA) and dihydrolipoic acid (DHLA). The experiment were conducted in PC12 and Caco-2 cells exposed to toxic metal (As, Cd and Pb).
The protection impact registered was related to the reduced cellular oxidative stress through enhancement of the antioxidant defense system.
To improve the manuscript, authors should consider the following recommendations:
- Authors must reconsider the quality of the figures provided.
- English language revision must be carried to improve the quality of the manuscript.
Reviewer 3 Report
1) you state
alpha-lipoic acid (ALA/LA)-why there are two abbreviations?
2) you state
dihydrolipoic acid (DHLA) are endogenous
what do you mean by endogenous? where?
3) replace
to reduce toxic metal
with
to reduce metal
4) what is meant exactly by cleaved PARP1?
5) in introduction you state
-lipoic acid (ALA/LA), one of the dithiol compounds often reduced enzymatically 33
to dihydrolipoic acid (DHLA), and gained much interest due to their potential role in 34
free radical scavenging, able in chelating metals and restoring intracellular glutathione 35
(GSH) levels when toxicants accretion, environmental insults and senescence were oc- 36
curred by environmental pollutants such as heavy metals [1-3]
this is an extremely long sentence and it does not make much sense in English. please shorten in smallet concise sentences so that the meaning is clear
6) you state
as an asserted antioxidant compound
what does asserted mean?
7) lines 40-42 please see comment 5-please ameliorate English
8) I dont think that via should be in italics
9) you state blood-brain barrier (BBB)
in case phrases are not used again please do not add abbreviations. If a phrase is abbreviated, then only the 1st time give the full phrase and after that use the abbeviation only
10) I am not sure that so much extensive information for LA and DHLA is needed in the introduction-maybe some of this should be in the discussion?
11) since these substances are so well studied what is the novelty of your study on them? please stress so in the introduction. also a concluding sentence in the introduction is needed
11) how relevant to real in vivo conditions of mammalian cells are the cell lines that you have used? please clarify
12) I do not understand very well your DNA damage/apoptosis assay. Is it described in Hossain et al? how exactly you differentiated between DNA damage and cell apoptosis? for example for DNA damage (not apoptosis) the comet assay is used. I am not sure how all these were calculated (Fig 6 is not very informative for me). I would appreciate some in depth analysis on this
13) I do not understand the statistics you performed. I believe you did ANOVA and then multiple post hoc comparisons as such why there are both asterisks and letters? this makes no sense. please just show different letters when there are differences. if they are different to control, just show the difference to controls. you also have to state in materials and methods the statistics not only show them in the results
14) how exactly you measured the quantitative difference in the western blot?
15) you state Despite the pro-oxidant potential at high dose level of -lipoic acid (LA) at low 298
dose level it has antioxidant role as well in oxidative stress condition
this sentence is not understandable
16) the discussion needs to be rewritten with the help of a fluent English speaker also if it could be shorter. also please if it could be clearer becaure right now its not appealing to the reader
17) please make sure that the references are written according to instructions for authors
Reviewer 4 Report
The manuscript describes the effect of lipoic acid (LA) and dihydrolipoic acid (DHLA) on cancer cells (PC12, Caco-2) treated with toxic heavy metals and their potential application in detoxifying heavy metal cations. It was proved in this study that LA and DHLA decreases apoptosis of cancer cells (PC12, Caco-2) induced by As, Cd, or Pb cations, presumably due to their antioxidant activity. However, in most datasets, the results of single-concentration experiments are provided, and the working concentration is quite high (250 microM). To clearly demonstrate the effect of heavy metal cations on cancer cells and the effect of LA (DHLA) against heavy-metal-induced cell damage (cell viability, GSH level), the authors need to provide the concentration-response curve (Refer to W.G. Carter’s ‘Plos one’ paper (2019) for the effect of heavy metals on cell viability and M.J.Nam’s ‘Human and experimental toxicology’ paper (2019) for the effect of LA (DHLA)).
It is not persuasive to claim the effect of LA(DHLA) on genotoxicity caused by heavy metals based on the DNA electrophoresis results in Figure 6B (without any explanations), and DNA density (Figure 6C). More experimental evidence is needed, including what the authors mentioned ‘data not shown’.
In the Weston blot analysis depicted in Figure 7, some bands (including that for control) are too thick. The authors should reexamine the experimental condition to get the image of clear bands for comparison.
Thorough English proof reading is required for the entire manuscript. The figures need to be revised as each figure has different size, formats and text(font/size).
Round 2
Reviewer 1 Report
The authors replied to most of the review comments in a satisfactory and reasonable way.
However, there are two points for authors to consider, as follows.
1) Line 56. C12 cells (neuron-like rat brain cells), which are well-known and “popular” as a model cell line for toxicity assessment…
To use the word “commonly used” would be better in terms of scientific writing.
2) Most of the whole pictures of the immunoblot provided by the authors are of reasonable quality. However, the image quality of the immunoblot for Nrf2 (Figure 9A) is quite poor. There are some cutting lines between bands. To avoid the risk of misunderstanding, please replace this blot of Nrf2 with a new blot.
Reviewer 3 Report
The manuscript is very much improved
I suggest you replace one and two asterisks with a and b or any other symbols because one asterisk usually means p<0.05 and two asterisks mean p<0.01
please proofread by a fluent English speaker and maybe ask the aid of the journal eg some mistakes in abstract
replace
Both significantly relieved the cells from Cd (5 M), As (5 M), Pb (5 M)-induced cell death
with
Both significantly decreased Cd (5 M), As (5 M), Pb (5 M)-induced cell death
replace
Subsequently, Both ALA 20
and DHLA improved the cell membrane integrity and recovered intracellular glutathione (GSH) 21
level, from metal-induced toxicity
with
Subsequently, Both ALA 20
and DHLA restored cell membrane integrity and intracellular glutathione (GSH) levels, which has been affected by metal-induced toxicity
replace
In addition, DHLA protected PC12 cells from metals-induced 22
DNA damage upon co-exposure. Furthermore, ALA and DHLA upregulated the expression of 23
survival-related proteins mTOR, Akt, Nrf2 in PC12 cells, that were downregulated by metal- 24
exposure.
with
In addition, DHLA protected PC12 cells from metal-induced
DNA damage upon their co-exposure to metals. Furthermore, ALA and DHLA upregulated the expression of
survival-related proteins mTOR, Akt, Nrf2 in PC12 cells, that had previously been downregulated by metal- 24
exposure.
replace
In contrast, in Caco-2 cells upon co-exposure using metals and ALA, Nrf2 was upregu- 25
lated and cleaved PARP-1 was downregulated. These findings suggest that ALA and DHLA are 26
capable of reducing the toxic effects of metals. The protection may be largely due to an enhance- 27
ment of antioxidant defense which reduces glutathione level and reduces the cellular oxidative 28
stress
with
In contrast, in Caco-2 cells upon co-exposure to metals and ALA, Nrf2 was upregulated and cleaved PARP-1 was downregulated. These findings suggest that ALA and DHLA can counterbalance the toxic effects of metals. T
you state
The protection may be largely due to an enhance- 27
ment of antioxidant defense which reduces glutathione level and reduces the cellular oxidative 28
stress
I dont understand this sentence-what reduces GSH the metals or the enhancement of the defense? what reduces oxidative stress? the GHS reduction or the enhancement of defense? please rephrase
Reviewer 4 Report
The manuscript has been improved a lot, especially for those I have recommended in the first-round peer review. I suggest the publication of this manuscript.
